# Structural basis of ALC1/CHD1L autoinhibition and the mechanism of activation by the nucleosome

Li Wang[1,2,4], Kangjing Chen[1,2,4] & Zhucheng Chen ✉[1,2,3]

Chromatin remodeler ALC1 (amplification in liver cancer 1) is crucial for repairing damaged DNA. It is autoinhibited and activated by nucleosomal epitopes. However, the mechanisms by which ALC1 is regulated remain unclear. Here we report the crystal structure of human ALC1 and the cryoEM structure bound to the nucleosome. The structure shows the macro domain of ALC1 binds to lobe 2 of the ATPase motor, sequestering two elements for nucleosome recognition, explaining the autoinhibition mechanism of the enzyme. The H4 tail competes with the macro domain for lobe 2-binding, explaining the requirement for this nucleosomal epitope for ALC1 activation. A dual-arginine-anchor motif of ALC1 recognizes the acidic pocket of the nucleosome, which is critical for chromatin remodeling in vitro. Together, our findings illustrate the structures of ALC1 and shed light on its regulation mechanisms, paving the way for the discovery of drugs targeting ALC1 for the treatment of cancer.

[1] Key Laboratory for Protein Sciences of Ministry of Education, School of Life Science, Tsinghua University, 100084 Beijing, P.R. China. [2] Beijing Advanced Innovation Center for Structural Biology & Beijing Frontier Research Center for Biological Structure, School of Life Science, Tsinghua University, 100084 Beijing, China. [3] Tsinghua-Peking Center for Life Sciences, 100084 Beijing, China. [4] These authors contributed equally: Li Wang, Kangjing Chen.
✉email: zhucheng_chen@tsinghua.edu.cn

Packaging the genome into chromatin within the nucleus blocks access to the DNA. Chromatin remodelers alter the positions and compositions of nucleosomes, regulating the chromatin structure and nuclear transactions[1]. Tight control of the activity of remodeling enzymes ensures the proper chromatin landscape and cellular functions. ALC1, also known as CHD1L (chromodomain helicase/ATPase DNA-binding protein 1-like), is an ATP-dependent chromatin remodeler that relaxes chromatin and plays an important role in the poly(ADP-ribose) polymerase 1 (PARP1)-mediated DNA repair pathway[2–6]. Defect in ALC1 regulation is associated with the development of hepatocellular carcinoma (HCC)[2,3,7].

ALC1 contains two RecA-like ATPase domains (lobe 1 and lobe 2, Fig. 1a) and a C-terminal macro domain. The macro domain binds to the ATPase motor and inhibits the enzyme in the nucleosome-free state[8,9]. Mutations of several basic residues on the macro domain, which were cancer-related, were found to disrupt ALC1 autoinhibition, compromising its function in cells. Whereas hydrogen-deuterium exchange (HDX) and mutagenesis studies suggest that the macro domain interacts with lobe 2, small-angle x-ray scattering (SAXS), and cross-linking mass-spectrometry (XL-MS) suggest that ALC1 is in mixed conformations, with the macro domain binding either of the RecA-like domains[8,9].

Upon DNA damage, PARP1 is activated and catalyzes the synthesis of poly(ADP-ribose) (PAR) chains on itself and on many other proteins[10]. The PAR chains bind to the macro domain of ALC1, releasing the autoinhibition of ALC1 and targeting the remodeler to the sites of DNA damage, where ALC1 slides the nucleosome and opens the chromatin[2,3,8]. ALC1 activation also requires the histone H4 tails of the nucleosome[2]. The H4 tails and other nucleosomal epitopes often provide the combinational cues to target and regulate chromatin remodelers[11]. The loss or inactivation of ALC1 in cells results in sensitivity to DNA damages[2].

ALC1 is an attractive target for cancer treatments. ALC1 is amplified in over 50% of cases of HCC, one of the commonest human cancers, which has a very poor prognosis and resistance to chemotherapy[12]. The oncogenic functions of ALC1, including its roles in enhanced cell motility, anti-apoptosis, accelerated mitotic progression, and tumor dedifferentiation, have been widely demonstrated both in vitro and in vivo[12–19]. ALC1 knockdown was shown to reverse tumor differentiation, abolish the malignant phenotypes, and increase the sensitivity of the HCC cells to chemotherapy. Small-molecule inhibitors of ALC1 exert potent antitumor activity by inhibiting the ATPase activity in colorectal cancer models[20]. ALC1 is indirectly blocked by PARP1 inhibitors, which have been used successfully in cancer treatment[10], suggesting that the clinical benefits of these drugs may be partially attributable to their indirect effects on ALC1.

Despite the importance in DNA repair and clinical implication, the structure of ALC1 and its regulation mechanisms remain unclear. To gain insights into the mechanisms of ALC1 autoinhibition and activation, we determined the crystal structure of human ALC1 and the cryoEM structure of its complex with the nucleosome.

## Results

**Structural determination of ALC1.** We determined the crystal structure of a human ALC1 construct containing the motor domain, the linker region and the macro domain in the presence of ADP-Mg (Fig. 1a). ALC1 is notoriously flexible, and crystallization was achieved with single-chain antibodies (scFvs), which were obtained by screening a yeast-display library. Several antibodies directed against ALC1 were isolated and one generated crystals diffracting to 3.5 Å (Suppl. Table 1). The antibody bound to lobes 1 and 2 of ALC1 (Fig. 1b), with no interaction with the key auto-inhibitory macro-lobe 2 interface (more discussion below), probably functioning as a rigid scaffold to fix the lobe 1-lobe 2 orientation and hence facilitating crystallization.

The structure shows that lobes 1 and 2 of ALC1 have little direct interaction (Fig. 1b and Suppl. Fig. 1a), suggesting a large degree of conformational plasticity. This is consistent with the multiple conformations of the enzyme observed by negative strain EM[9], explaining the difficulty of crystallizing ALC1 in the absence of the antibody. The individual lobes 1 and 2 of ALC1 adopt a typical RecA-like fold[21], showing local structural rearrangement of lobe 2 because of the binding to the macro domain (Suppl. Fig. 1b, c). The linker region is largely disordered, with a short loop region binding to lobe 2 and the preceding segment forming a helix that packs against a nearby molecule in the crystals (Suppl. Fig. 1d).

**Mechanism of ALC1 autoinhibition.** The macro domain interacts with lobe 2 and sequesters two elements that are important for nucleosome recognition (Fig. 2a and Suppl. Fig. 2), which provides the structural basis of ALC1 autoinhibition. The macro domain of ALC1 adopts a highly conserved macro domain fold, with Dali Z-scores of 14.7 and 10.2 compared with the macro domains of DarG and macroH2A1.1, respectively[22,23] (Suppl. Fig. 3). The macro domain of ALC1 binds to a saddle-shaped surface of lobe 2 formed by the conserved helicase motif IV and the H4-binding sites, with the ADP-ribose binding pocket exposed to the solvent. Lobe 2-macro domain binding occurs through mixed hydrophobic and H-bond interactions, covering an area of ~750 Å².

The β5-α4 loop, referred to as the P-loop (because the equivalent sequences in canonical macro domains bind to the phosphate group of ADP-ribose (Suppl. Fig. 3)), adopts an extended conformation and interacts with the long side chain of Arg402 of motif IV (Fig. 2b). The side chain of Tyr874 (the equivalent residues in DarG and macroH2A1.1 pack against the adenosine ring of ADP-ribose (Suppl. Fig. 3)) interacts with Ser396 (Fig. 2b). The bulky side of Trp852 of the macro domain is buried by the binding interface, contacting Arg402 and Arg398 of motif IV. In support of the structure, relative to the protein with

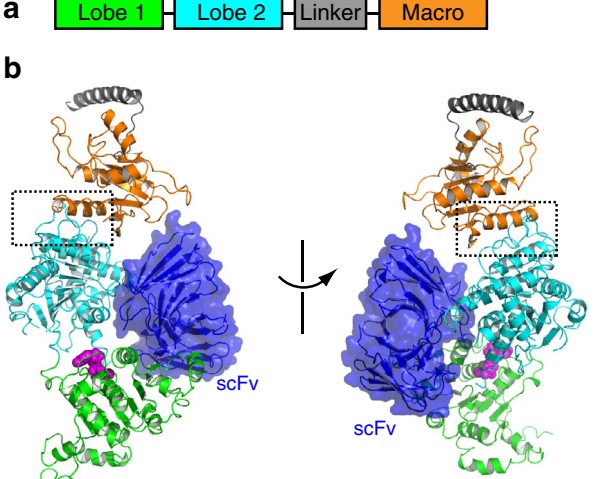

**Fig. 1 Structure of ALC1 in the autoinhibited state. a** Domain architecture of ALC1. **b** Two different views of the crystal structure of ALC1 in the autoinhibited state. Bound nucleotide are colored magenta spheres. The boxed region is enlarged for analysis in Fig. 2. The bound scFv is colored blue.

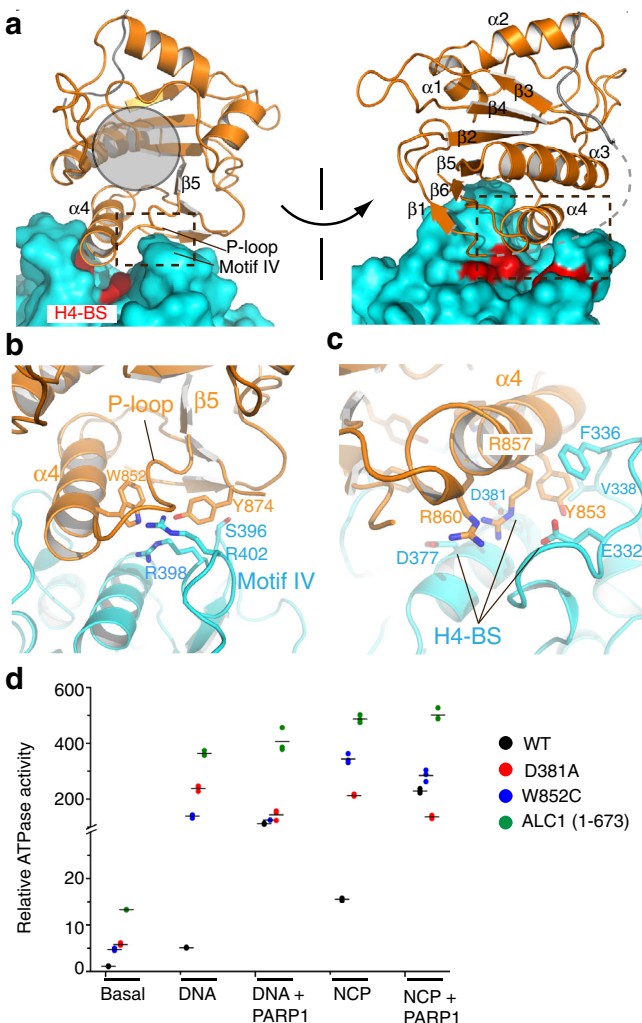

**Fig. 2 Mechanisms of ALC1 autoinhibition. a** Two different views of the macro-lobe 2 interaction. Lobe 2 is shown as a surface model. The position of the putative ADP-ribose binding pocket is indicated with a gray circle. Three acidic residues of the H4-binding surface (H4-BS) are colored red. The boxed regions are enlarged and analyzed in **b**, **c**. **b** Interaction between the macro domain and motif IV. **c** Interaction between the macro domain and the H4-binding surface. **d** ATPase activities of ALC1 and three mutants under different conditions. Basal: activity without DNA, NCP, or PARP1. All data points are showed with lines indicating the means ($n = 3$).

wild-type interface (WT), the W852C mutant, which is associated with cancer[24], increased the basal and DNA-dependent ATPase activities by factors of 7 and 35, respectively (Fig. 2d and Suppl. Fig. 4). Unlike the WT protein, PARP1 did not further activate the mutant. The data supported that the macro-motif IV interaction is essential for ALC1 inhibition and the W852C mutation hyperactivates ALC1, bypassing the requirement for the PAR chain. This observed binding interface of motif IV is consistent with the HDX2 region as detected previously using HDX assays[8]. Helicase motifs IV of Snf2-like chromatin remodelers typically adopt a helical conformation, and bind to the nucleosomal DNA[21]. In the structure of ALC1, motif IV adopts a loop conformation, in which the DNA-binding residue Arg402 is buried by the interaction with the macro domain, and is thus inaccessible to the nucleosome substrate. The sequestra-tion of DNA-binding motif IV provides one rationale for the autoinhibition of ALC1.

The macro domain engages lobe 2 through a second element, the H4-binding surface (Fig. 2c). Arg857 and Arg860 at the C-terminus of α4 form H-bonds and charge–charge interactions with Glu332, Asp377, and Asp381. Tyr853 at the N-terminus of α4 makes hydrophobic contacts with the side chains of Phe336 and Val338 of lobe 2. In support of the structure, the D381A mutation notably increased the PARP1-independent ATPase activities (Fig. 2d). This structure is also supported by the cancer-associated mutations at Arg857 and Arg860, which release the autoinhibition of ALC1[8,9]. This macro-domain-binding interface is in line with the HDX1 region detected with the H/D exchange assays[8]. Interestingly, the acidic pocket formed by Glu332, Asp377, and Asp381 of ALC1 is similar to those of ISWI and Snf2 (Suppl. Fig. 1c), which bind to the H4 tails[21,25]. The H4 tails are important for ALC1 activation[2]. The structure suggests that the macro domain sequesters the H4-binding sites, which provides a second mechanism of ALC1 autoinhibition.

Compared with the WT protein under the same conditions, deletion of the macro domain in ALC1 (1-673) dramatically stimulated the basal and DNA-dependent ATPase activities by factors of ~20 and ~120 (Fig. 2d), respectively, which are much higher than the activities caused by the individual mutations of motif IV and the H4-binding interfaces, suggesting these two elements cooperate to repress the activity of ALC1. Together, the macro domain of ALC1 uses two strategies to inhibit the motor, combining its interactions with the DNA-binding motif IV and the H4-binding surface to tightly suppress the enzyme in the ground state. Several elements within the ADP-ribose binding pocket of the macro domain, the P-loop in particular, are directly involved in the sequestration of lobe 2, suggesting that PAR chain binding triggers conformational changes in the macro domain, releasing motif IV and the H4-binding sites to recognize the nucleosome in the activated state.

**Structure of ALC1 bound to the nucleosome.** To gain insights into ALC1 activation by the nucleosome, we determined the cryoEM structure of the R857Q mutant bound to the nucleosome in the presence of a stable ATP analog, ADP-BeF$_x$. The R857Q mutant is known to release the autoinhibition of ALC1, bypassing the requirement for the PAR chain[8,9]. The structure of the ALC1-nucleosome complex was determined at an overall resolution of 2.8 Å, with local resolutions of 2. 8 Å and 3.1 Å at the nucleosome bound with the ALC1 linker and the ALC1 motor domain, respectively (Suppl. Figs. 5 and 6 and Suppl. Table 2). ALC1 bound to the nucleosome at superhelical location 2 (SHL 2), in a manner conserved with Snf2 and other chromatin remodelers[26] (Fig. 3a and Suppl. Fig. 7a). Lobe 1 and lobe 2 of ALC1 underwent large structural transitions, with the ATP-binding/hydrolysis motifs I and VI realigning in close proximity to one another (Suppl. Fig. 7b). The brace helix of lobe 2 extended one more helical turn and interacted with lobe 1 (Suppl. Fig. 7c). These conformational changes are consistent with ALC1 in the activated state. In the cryoEM samples, some nucleosomes were bound by two copies of ALC1 symmetrically, the structure of which was refined to 3.3 Å (Suppl. Figs. 5c, 6i, and 7d). In this double-binding model, ALC1 interacted with the nucleosome in a manner similar to that of the high resolution single-binding structure (Suppl. Fig. 7e), which is the focus of this study.

In the activated state, the macro domain dislodged from lobe 2 and its structure could not be resolved, suggesting that the macro domain did not bind stably to the nucleosome or the ATPase motor domains. This is in line with the previous findings that the macro domain is not essential for chromatin relaxation in cells[8]. Consistent with this notion, the macro domain deletion mutant ALC1 (1-673) showed a greater remodeling activity than the WT

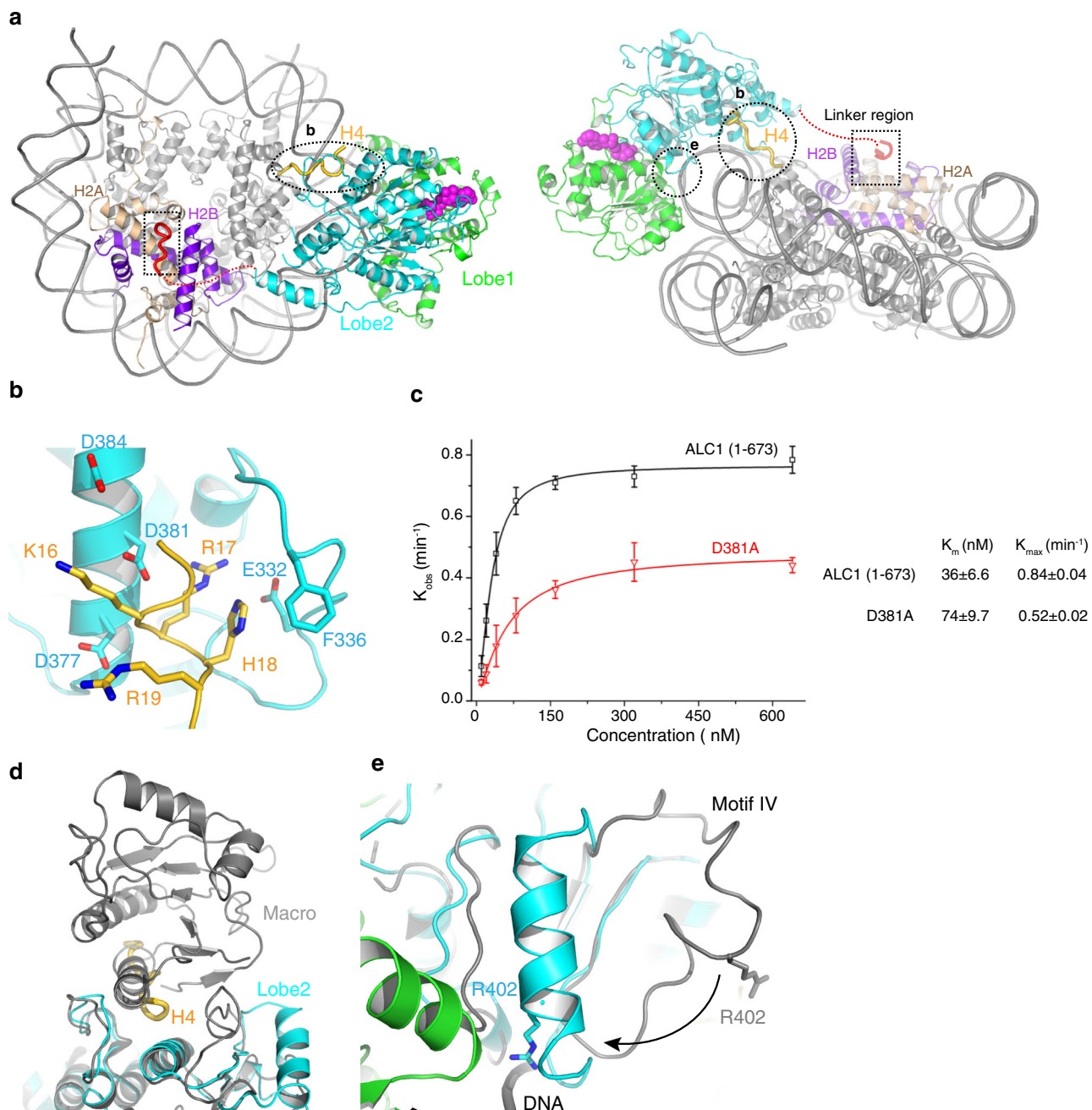

**Fig. 3 Mechanisms of ALC1 activation by the nucleosome. a** Top and front views of the ALC1-nucleosome complex. The linker region of ALC1 bound to the surface of the nucleosome is colored red, and enlarged for analyses in Fig. 4; the H4 tail bound to ALC1 is colored gold. The encircled regions are enlarged for analyses in **b** and **e**. **b** Interaction between ALC1 and the H4 tail. **c** The observed remodeling rates of ALC1 (1-673) and D381A mutant at multiple enzyme concentrations. The data were fit with a hyperbola function, and the resulted $K_m$ and $K_{max}$ were shown on the right. Error bars indicate standard deviations for three independent measurements and the measure of center for the error bars is the mean value. **d, e** Structural comparisons of the autoinhibited (colored gray) and nucleosome-bound (color coded) states, around the H4-binding surface (**d**) and motif IV (**e**). The structures of lobe 2 are aligned.

protein (Suppl. Fig. 8). Different from the WT protein, the remodeling activity of the mutant was, however, not further enhanced in the presence of PARP1. These data indicated that the macro domain inhibits the ATPase activity of ALC1 in the nucleosome-free state, but is not essential for the remodeling reaction in the activated state.

In line with the structures of Snf2 and ISWI[21,25], the acidic pocket of lobe 2 formed by Glu332, Asp377, Asp381, and Asp384 of ALC1 interacts with the basic patch $K_{16}R_{17}H_{18}R_{19}$ of the H4 tail of the nucleosome (Fig. 3b and Suppl. Fig. 6b). The breadth of

the pocket is increased by the hydrophobic packing of Phe336 of ALC1 to His18 of the H4 tail. In agreement with the requirement of H4 for ALC1 activation, disruption of the H4-binding interface by the D381A mutation diminished the remodeling activity (Suppl. Fig. 9a). Therefore, the functions of Asp381 are twofold. It interacts with the macro domain in the nucleosome-free state (Fig. 2c), inhibiting the enzyme in the absence of the substrate, whereas it interacts with H4 tail in the nucleosome-bound state, promoting the remodeling activity. To get insights into the loss of the remodeling activity caused by the D381A mutation at

the activated state, we performed the experiments at multiple enzyme/substrate concentrations to do a $K_m/K_{cat}$ analysis using the construct without the macro domain, which bypassed the complication due to the autoinhibition and requirement for PARP1 activation (Fig. 3c). The data showed that the D381A mutation increased the Km ~2-fold, while it slightly decreased the catalytic efficacy ($K_{max}$), suggesting the defect was mainly because of the compromised nucleosome binding. The competition of the H4 tails with the inhibitory macro domain for binding to the same surface of lobe 2 explains the requirement for this nucleosomal epitope for ALC1 activation (Fig. 3d). Likewise, motif IV underwent a loop-to-helix conformational change (Fig. 3e), with the side chain of Arg402 binding to the phosphate backbone of nucleosomal DNA, as typically observed in the activated state of chromatin remodelers[26]. Therefore, the release of the macro domain frees the H4-binding surface and motif IV to interact with the nucleosome, providing the mechanism of ALC1 activation.

### Recognition of the acidic pocket of H2A-H2B by ALC1.
Notably, we found that a segment of the linker region (residues 610–617) adjacent to lobe 2 binds to the H2A-H2B acidic patch of the nucleosome (Figs. 3a and 4a, b). In particular, two arginine residues, Arg611 and Arg614, play critical roles. Arg611 interacts with Gln44 of H2B and Glu56 of H2A, whereas Arg614 forms a network of H-bonds with Glu61, Asp90, and Glu92 of H2A. Arg611 and Arg614 are highly conserved in vertebrate (Suppl. Fig. 10), suggesting their functional importance. Mutations at Arg611 and Arg614 are found in cancer patients, with Arg611 mutations recurring in multiple cases[24].

We conducted microscale thermophoresis (MST) assays to determine the binding affinity of the linker region with the H2A-H2B dimer (Fig. 4c). Whereas the WT peptide bound to the H2A-H2B dimer with a disassociation constant Kd ~ 3.7 μM, the R611E and R614E mutation reduced the affinity over 10-fold, with Kds ~41 μM and 113 μM, respectively, supporting that both arginine residues are important for the anchor of ALC1 to the acidic patch of the nucleosome.

To test the function of the linker region-nucleosome interaction, we mutated the key arginine residues and measured the ATPase and remodeling activities in vitro. The mutations R611E and R614E did not alter the ATPase activity (Suppl. Fig. 9b), but caused notable losses of the remodeling activity (Fig. 4d and Suppl. Fig. 9c). The double mutant R611E/R614E showed a defect similar to that observed with the single mutants, suggesting that the two arginine residues work in a conjugated manner to facilitate the remodeling reaction.

The binding of Arg611 and Arg614 to the acidic pocket of the nucleosome is unexpected. We did the reciprocal experiment with mutations of the acidic patch to further validate these findings. Consistent with our model, disruption of the acidic patch reduced the remodeling activity of the WT enzyme(Fig. 4d and Suppl. Fig. 9d). Moreover, the R611E and R614E mutations were not worse on the acidic patch mutant nucleosomes. The data support that the arginine anchors of ALC1 and the acidic patch of the nucleosome work through the same mechanism to regulate the activity of the enzyme.

The loss of the remodeling activity was probably not because of thermal instability, as addition of more mutant protein at a later time point, and lowering the reaction temperature, did not

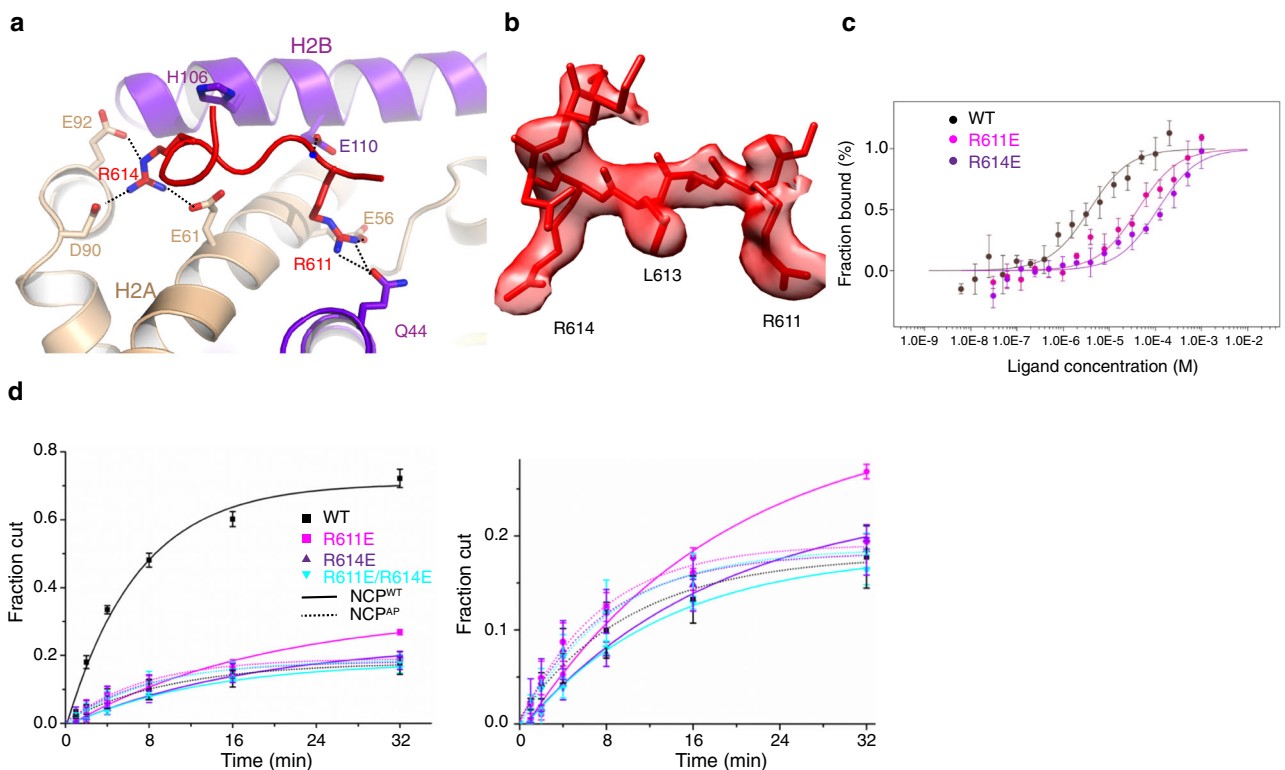

**Fig. 4 Binding of the linker region of ALC1 to the acidic patch of the nucleosome. a** Structure of the linker region of ALC1 bound to the acidic patch of the nucleosome. **b** Local cryoEM density around the linker region of ALC1. **c** MST measurements of the H2A-H2B binding affinities of the WT and mutant ALC1 peptides (residues 604–623). Error bars indicate standard deviations for three independent measurements and the measure of center for the error bars is the mean value. **d** Chromatin remodeling activities of WT and three mutant ALC1 toward the intact nucleosome (NCP$^{WT}$) and acidic patch mutant nucleosome (NCP$^{AP}$, dotted lines). Error bars indicate standard deviations for three independent measurements and the measure of center for the error bars is the mean value. The activities of the mutants are enlarged and shown on the right.

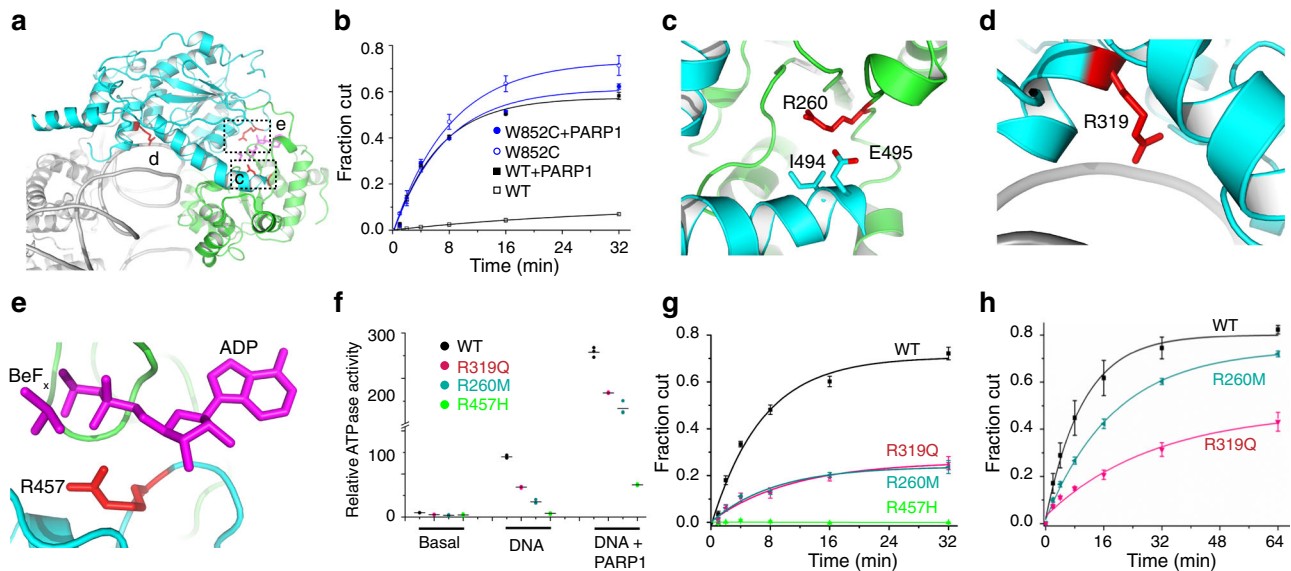

**Fig. 5 Dysregulation of ALC1 by mutations found in cancer patients. a** Map of cancer-associated mutations of ALC1 at the bioactive interfaces. Boxed regions are enlarged and analyzed in (**c–e**). **b** Chromatin remodeling activities of WT and W852C mutant in the presence (filled symbols) and absence (open symbols) of PARP1. Error bars indicate standard deviations for three independent measurements and the measure of center for the error bars is the mean value. **c–e** The mutations at Arg260, Arg319, and Arg457 map to the lobe 1-lobe 2 interface (**c**), the lobe 2-DNA-binding interface (**d**), and the catalytic interface for ATP hydrolysis (**e**), respectively. **f** ATPase activities of WT, R260M, R319Q, and R457H mutant ALC1. All data points are shown with lines indicating the means ($n = 3$). **g** Chromatin remodeling activities performed at 37 °C in the presence of PARP1. **h** Chromatin remodeling activities at 25 °C. Error bars indicate standard deviations for three independent measurements and the measure of center for the error bars is the mean value. Fitting the data indicated the remodeling rate constants were ~0.1 and ~0.05 min$^{-1}$ for the WT and the R260M mutant, respectively.

noticeably promote the reaction (Suppl. Fig. 9e). We noticed that the mutants failed to reach the same amount of remodeled nucleosome as the WT enzyme. A possibility is inhibition by the product of ATP hydrolysis (ADP).

**ALC1 dysregulation by cancer-associated mutations.** In addition to gene amplification, over 200 missense mutations of ALC1 are reported in the COSMIC database[24]. Many of these occur in bioactive regions, disturbing the varied mechanisms of ALC1 regulation (Fig. 5a). As discussed above (Fig. 2), the mutations at Trp852, Arg857, or Arg860 of the macro domain release the ALC1 autoinhibition. The Arg857 and Arg860 mutants were previously shown to be constitutively active[8,9], and we evaluated the impact of the W852C mutation on the chromatin remodeling activity, Fig. 5b. Consistent with the ATPase activity (Fig. 2d), the W852C mutant was hyperactive in the absence of PARP1, whereas it displayed an activity similar to WT in the presence of PARP1, suggesting Trp852 is important for autoinhibition of ALC1, and dispensable for the remodeling activity when the enzyme is activated by PARP1 in vitro.

Arg260 maps to lobe 1 and is in close contact with Ile494 and Glu495 of the brace helix at the lobe 1-lobe 2 binding interface (Fig. 5c); Arg319 is close to the nucleosomal DNA (Fig. 5d); and Arg457 is one of the "arginine fingers" that catalyze ATP hydrolysis (Fig. 5e). As expected, the catalytic R457H mutation abolished the ATPase (Fig. 5f) and remodeling activities (Fig. 5g and Suppl. Fig. 9f) in vitro. The cancer-associated mutations R260M and R319Q did not alter the ATPase activities much (Fig. 5f), but markedly diminished the remodeling activities. The R260M mutation seemed to perturb the thermal stability, as the mutant enzyme remodeled a higher amount of the nucleosome at a lower temperature, to a level similar to WT under the same conditions (Fig. 5h and Suppl. Fig. 9g). Fitting the data indicated that the remodeling rate constant of the mutant enzyme was about half of the value of the WT protein, suggesting the R260M

mutation impacts both the stability and remodeling efficiency of the enzyme. The data supported that the remodeling activity of ALC1 is regulated through multiple fashions, and our findings provide a framework to analyze the functions of the ALC1 mutants in cancer cells.

## Discussion

Our studies reveal how ALC1 is autoinhibited and provide the mechanisms by which it is regulated by the nucleosomal epitopes. Chromatin remodelers are often autoinhibited to avoid futile ATP hydrolysis in the absence of the nucleosome substrate. Snf2 is inhibited through direct stacking of the two RecA-like lobes[27]. The ATPase motor of Chd1 is inhibited by the N-terminal double chromodomain[28], which binds lobe 2 and sequesters the DNA-binding elements, motif V in particular, with the H4-binding surface exposed. ISWI is tightly autoinhibited by the AutoN domain, which sequesters motif V and the H4-binding surface[25]. The regulatory mode of ALC1 shows analogy to the auto-inhibitory mechanism of ISWI, in that the macro domain binds to lobe 2, inhibiting the enzyme by sequestering motif IV and the H4-binding surface. Different from ISWI, in which the AutoN domain remains binding to the lobe1 of the enzyme at the activated state[29], the macro domain of ALC1 dislodges from the motor when activated by the nucleosome. Disruption of the macro-lobe 2 interaction of ALC1 by cancer-associated mutations, or binding of the PAR chain, allows the enzyme to realign the ATPase lobes and engage the nucleosome. In the activated state, the H4 tails and the acidic patch of the nucleosome interact with lobe 2 and the linker region of ALC1, respectively.

The previous studies supported the importance of the linker sequence of ALC1 in cells[8]. Out data suggest that binding of the linker region through the arginine anchors to the H2A-H2B acidic pocket of the nucleosome stabilizes the activated conformation and/or prevents the enzyme from zipping back to the autoinhibited state, providing the mechanisms of ALC1

regulation by the nucleosomal epitopes. The residues neighboring the H2A-H2B acidic pocket are subjected to modifications, and the acidic pocket is believed to function as a tunable hub for many chromatin proteins[11,30]. Several chromatin remodelers, including Snf2 and ISWI, are also regulated by the H2A-H2B acidic pocket, although the mechanisms are not completely clear[31–33]. Poly-basic motifs are found in ISWI enzymes and the SnAc domain of Snf2. The interaction mode between the dual arginine "RxxR" motif of ALC1 and H2A-H2B of the nucleosome provides a framework for understanding the regulation mechanisms of other chromatin remodelers. While our manuscript was under review, a study by Lehmann et al. reported the structure of the isolated ALC1 linker peptide bound to the acidic patch of the nucleosome[34], which, however, shows an interaction mode different from our structure. The Lehmann's structure revealed only one of the arginine anchors. More importantly, the isolated ALC1 linker peptide bound to the nucleosome with Arg611 interacting Glu64, Glu61, Asp90, and Glu92 of H2A, whereas Arg614 bound to this acidic pocket in our structure. We noticed that the EM density, which fit neatly to the side chain of Leu613 in our structure (Fig. 4b), could not easily be explained by the smaller side chain of Ser612 in the Lehmann's structure.

ALC1 is regulated by diverse mechanisms, for example, by modulating macro inhibition, disrupting the lobe 1-lobe 2 interaction, perturbing H4 binding, and interfering ATP hydrolysis. Tight regulation of ALC1 action, chromatin remodelers in general, is critical for the function in vivo. We speculate that the constitutively active mutants, such as W852C and R857Q, probably excessively relaxes the chromatin even in the absence of DNA damage, leading to oncogenic activities as found during ALC1 overexpression. In contrast, mutations of Arg611 and Arg614 cause loss of ALC1 function, which would sensitize cells to DNA damages in a manner similar to that seen in the ALC1-deficient cells[2]. More studies are needed to understand the in vivo effects. ALC1 is a promising target for cancer treatment, and small-molecule inhibitors targeting the ATPase activity have been found, although they showed low affinities[20]. In addition to the common strategy of inhibiting the ATPase activity, the sensitivity of the remodeling activity and the varied regulation modes of ALC1 provide rich opportunities and a large chemical space to control the function of this enzyme in cells. The structures and regulation mechanisms revealed in this study pave the way for the future discovery of drugs that target ALC1 in cancer treatments.

## Methods

**ALC1 expression and purification.** The gene of ALC1 (residues 1–880) was cloned from Homo sapiens complementary DNA (cDNA), and inserted into a modified pET-28b vector containing a SUMO tag. The ALC1 mutants were generated through Quickchange mutagenesis or Gibson assembly, and confirmed by DNA sequencing.

ALC1 was overexpressed in the Escherichia coli expression strain Rosetta (DE3). Cells were grown in LB media at 37 °C until the absorbance at 600 nm (OD$_{600}$) reached ~0.8. The temperature of the culture was reduced to 16 °C before addition of 0.5 mM isopropyl-beta-D-thiogalactoside (IPTG). Cells were cultured overnight, and harvested by centrifugation at 4000 r.p.m. (Beckman, Rotor JA10) for 15 min and resuspended in 20 mM HEPES, 1 M NaCl, 5% glycerol, 0.1% NP-40, 2 mM benzamidine, and 1 mM phenylmethanesulfonyl fluoride, pH 7.5. Cells were disrupted by a high-pressure homogenizer (ATS). After high-speed centrifugation, the supernatant was loaded on a Ni–NTA column and allowed to flow through by gravity. After a washing step with 25 mM imidazole, the protein was eluted with 20 mM HEPES, 150 mM NaCl and 250 mM imidazole, pH 7.5. The His6-SUMO tag was cleaved by Ulp at 4 °C overnight. The protein was further purified on an ion-exchange column (Source-15S, GE Healthcare), and then subjected to gel-filtration chromatography (Superdex-200, GE Healthcare) in buffer containing 10 mM HEPES, 150 mM NaCl and 10 mM dithiothreitol (DTT), pH 7.5. The purified protein was concentrated to 13 mg/ml and stored at −80 °C. The ALC1 mutants were purified similarly, and concentrated to 5–10 mg/ml.

We cloned the gene of PARP1 from Homo sapiens cDNA, and inserted it into a modified pET-28b vector. The construct was confirmed by DNA sequencing. PARP1 was overexpressed and purified using the similar protocol as described above. PARP1 was concentrated to 10 mg/ml for assays. All primers used were provided in Suppl. Table 3.

**Single-chain antibodies screening and purification.** Single-chain variable fragments (scFv) against ALC1 were obtained from a yeast-display library carrying non-immunized human scFv using a protocol as described before[35,36]. Briefly, two purified ALC1 proteins with different tags, ALC1-6His and ALC1-biotin, were used as antigens for screening the yeast-display library. ALC1-6His was constructed and inserted into a modified pET-28b vector, and purified similarly as described above. ALC1-biotin was obtained from coexpresseion of ALC1-avi (in a pET-28b vector) and BirA enzyme (in a pACYC vector) in Escherichia coli BL21 (DE3) strain. Streptavidin microbeads (200 μl, Miltenyi Biotec) were mixed with ALC1-biotin (100 nM) and 1 × 10$^{10}$ yeast cells in 5 ml PBS, and the bound candidate scFv cells were selected through a LS column, and then were amplified. In the sequential sorting, ALC1-6his pre-mixed with Anti-His-Biotin was used as the antigen for screening using Anti-Biotin microbeads in a similar way.

The isolated candidate scFv cells were then used for three rounds of flow cytometry sorting. In the first round of sorting, 100 nM biotinylated ALC1 was mixed with 2 × 10$^7$ candidate scFv cells and the relevant antibodies (chicken anti-c-Myc IgY and Alexa Fluor 488-goat anti-chicken IgG; streptavidin-allophycocyanin) in a final volume of 0.5 ml PBS. The candidate clones with a double positive population were isolated. In the next round of flow cytometry sorting, 6his-tagged ALC1 (chicken anti-c-Myc IgY and Alexa Fluor 488-goat anti-chicken IgG; streptavidin, R-phycoerythrin) was used. The third round of sorting was performed as the first round. After three rounds of sorting, the positive rate of isolated scFv clones was over 50%. Over 300 clones were confirmed by DNA sequencing. Finally, seven independent clones were identified on IgGblast website. All of the antibodies were purchased from Ivitrogen.

The seven genes of scFvs were cloned, and inserted into the pVRC-8400 vector, which contains an N-terminal signal peptide (MGWSCIILFLVATATCVHS) for secretion and a C-terminal 6×His tag for purification. The scFv plasmids were transfected into HEK293F cells (2 × 10$^6$ cells per ml) by PEI at a ratio 1:3. After 72 h transfection, the supernatant of cell culture containing the secreted scFv was collected, concentrated and buffer-exchanged to 20 mM HEPES, 500 mM NaCl,2 mM benzamidine, pH 7.5. The supernatant was loaded on a Ni–NTA column and allowed to flow through by gravity. After a washing step, the protein was eluted with 20 mM HEPES, 150 mM NaCl and 250 mM imidazole, pH 7.5. The elusion was further purified by gel-filtration chromatography using a Superdex 75 column (GE Healthcare) pre-equilibrated with elution buffer without the 250 mM imidazole. Fractions containing scFv were collected and then concentrated to 1 mg/ml and stored at −80 °C.

The purified scFv and ALC1 were incubated on ice for 1 h at a ratio of 1.1:1 in the binding buffer 20 mM HEPES, 150 mM NaCl, 10 mM DTT, pH 7.5. The mixture was purified by gel-filtration chromatography using a Superdex 200 column (GE Healthcare) pre-equilibrated with the same binding buffer. Fractions containing the complex were collected and were concentrated to 12 mg/ml and stored at −80 °C.

**Crystallization and data collection.** Crystals of ALC1-scfv complex were grown at 4 °C by hanging drop vapor diffusion above a reservoir solution from 35% (v/v)2-methyl-2,4-pentanediol (MPD), 100 mM acetate (pH 4.5), 10 mM DTT, with equal volumes of protein and reservoir buffer. Crystals were optimized with buffer 20% MPD, 100 mM HEPES (pH 7.0), 10 mM DTT, 0.5 mM ADP, 2 mM MgCl$_2$, and harvested in cryo-protectant containing extend 5–15% MPD and then flash-frozen in liquid nitrogen.

Diffraction data from crystals were collected at −170 °C at the beamline BL17U of Shanghai Synchrotron Radiation Facility.

**Data processing and structure solution.** The data were processed with HKL2000 suit[37]. The structure of ALC1-scfv complex was solved by molecular replacement using lobe 1 of Chd1 (Protein Data Bank accession number 3MWY)[28], lobe 2 of ISWI (Protein Data Bank accession number 5JXR)[25], macro domain (Protein Data Bank accession number 2FG1), and VH and VL domain of a human neutralizing antibody (5YAX) as the initial searching models[38]. The rest of the model was built manually using Coot[39]. Refinement was performed with Phenix[40]. The final structure was refined to 3.5 Å, with $R_{work}/R_{free} = 0.268/0.316$, Ramachandran outlier 0.10%, allowed 8.84%, and favored 91.06%.

**ATPase assays.** The ATPase activities were measured with a EnzChek Phosphate Assay Kit. The assays were performed with 0.1 μM ALC1 in the buffer of 3 mM ATP, 50 mM Tris-HCl (pH 7.5), 1 mM MgCl$_2$, 0.2 mM sodium azide, 0.2 mM DTT, and 50 mM sodium chloride. When indicated, 0.8 μM dsDNA (167 bp), 0.8 μM 167NCP, and 0.1 μM PARP1 and 25 μM NAD (pre-incubated with PARP1 at 37°C for 5 min) were used. To ensure the ATP substrate was not depleted, the linear fractions of the reactions, the early time points (the first 100 s in Fig. 2d, 300 s in Fig. 5f and Suppl. Fig. 9b, and 1200 s in basal reactions), were used to compare the activities of varied proteins. The specific ATPase activities of all the proteins were normalized to the basal activity of ALC1 in the absence of DNA and PARP1.

**Nucleosome remodeling assays**. Mononucleosome restriction enzyme accessibility assays were performed as described before[21]. Cy5-labeled mononucleosome (5 nM) and 0.1 μM of various ALC1 proteins (0.04 μM proteins in Suppl. Fig. 8) were incubated at 37 °C with 100 U of HhaI in the remodeling buffer (20 mM HEPES, pH 7.5, 50 mM NaCl, 5 mM MgCl₂, 5% glycerol, and 0.1 mg/ml bovine serum albumin). After adding 0.1 μM PARP1 (pre-incubated with 50 μM NAD at 37 °C for 5 min), 3 mM ATP was added to initiate the reaction. Fractions were taken at various time points and quenched with 2× Stop buffer (20 mM Tris, pH 8.0, 1.2% sodium dodecyl sulfate (SDS), 80 mM EDTA, and 0.2 mg/ml proteinase K). The reaction mixtures were incubated at 55 °C for 20 min to deproteinate the samples. The remodeling activities of ALC1 proteins (WT, R611E, R614E, R611E/R614E) towards the mutant NCP assembled with H2A (E56K/E61K/D90R/E92K) were obtained in the same way. To analyze the temperature dependency/thermal stability, the reactions were performed at 25 °C. To analyze the thermal stability of R611E, extra R611E (0.1 μM) was added at 32 min and reaction continued for another 32 min.

Fractions were running on 8% native TBE polyacrylamide gels in 0.25× TBE for 100 min at 120 V on ice. Gels were imaged using a Typhoon FLA9500 variable mode imager (GE Healthcare). Band intensities were quantified in Quantity One software. The reaction rate constants were fit to a single exponential decay using GraphPad Prism 8.4.3.

To compare the values of $K_m$ and $K_{max}$ of ALC1(1-673) and mutant D381A, different concentrations (10–640 nM) were titrated into the reaction mixtures. The measured $K_{obs}$ were fit to a Hyperbl model (OriginPro 8.5).

**Microscale thermophoresis**. MST analysis was performed using a Nano Temper Monolith NT.115 instrument (NanoTemper Technologies GmbH). H2A-H2B dimer was reconstituted as before[41]. It was fluorescently labeled with the RED-NHS 2nd Generation kit (NanoTemper Technologies GmbH), and desalted to the reaction buffer (10 mM MES, 175 mM NaCl, 0.05% Tween 20, pH 6.0). Peptides (WT(604-623), R611E and R614E) purchased from Scilight Biotechnology (95% purity), were dissolved to the reaction buffer and titrated to 16 different serial concentrations. The labeled dimer (50 nM) was mixed with the equal volume of unlabeled peptides before being loaded into standard glass capillaries (Monolith NT.115 Capillaries) at room temperature. The difference of the thermophoretic properties was measured at 649 nm with the laser power 40% (20% when the WT peptide was used) and medium MST power with the MO. Control software. The dissociation constant was measured with the MO. Affinity Analysis software. Each experiment was repeated three times.

**CryoEM sample preparation and data collection**. The nucleosome core particles (167NCPs) were constituted with 167 bp DNA containing the '601' positioning sequence (5'-strand:

CGCGGCCGCCCTGGAGAATCCCGGTGCCGAGGCCGCTCAATTGGTCG TAGACAGCTCTAGCACCGCTTAAACGCACGTACGCGCTGTCCCCGCGT TTTAACCGCCAAGGGGATTACTCCCTAGTCTCCAGGCACGTGTCAGATA TATACATCCTGAAGCTTGTCGA; the '601' positioning sequence is underlined) as described before[21].

The ALC1-NCP complex were obtained by mixing R857Q of ALC1 (30 μM) with 167NCP (6 μM). The complex was purified and stabilized using the GraFix protocol similarly as described before[21].

Quantifoil gold R2/1 grids with 200 mesh size were glow discharged in air for 30 s using a PDC-32G-2 Plasma Cleaner set to a low power. Samples (4 μl) were blotted for 3.5 s at −2 force before being plunge-frozen in liquid ethane with a FEI Vitrobot IV at 8 °C and 100% humidity.

Grids were examined and screened using an FEI Tecnai Arctica operated at 200 kV.

Cryo-EM data were collected using a Thermo Fisher Scientific Krios G3i operated at 300 kV equipped with a K3 direct electron detector and GIF Quantum energy filter (Gatan), at a magnification of ×81,000 for a final pixel size of 0.54125 Å/pixel in the super-resolution mode with the defocus values ranging from −1.3 to −1.8 μm. The total electron dose was 50 e−/Å² fractionated in 32 frames (exposure time 2.56 s).

**Image processing and model building**. A total of 8279 dose-fractionated image stacks were firstly aligned using MotionCor2[42] with twofold binning, resulting in a pixel size of 1.0825 Å/pixel. CTF parameters were estimated using CTFFIND4[43]. Particle picking, two-dimensional (2D) classification and three-dimensional (3D) classification were carried out in Relion3.0[44]. The initial picked particles were extracted with fourfold binning to increase signal to noise ratio, resulting in a pixel size of 4.33 Å/pixel. After 2 rounds of 2D classification, a total of 4,655,421 particles were selected and subjected to further processing. The ISWI-NCP complex (EMD-9718) were low passed to 60 Å as the initial model[29]. Particles were then re-extracted with twofold binning and after 3D classification a set of 1,102,628 particles with clear features were selected and yielded a structure of ALC1-NCP at an overall resolution of 4.4 Å. The particles were re-extracted without binning and subjected to CTF refinement. The overall resolution of ALC1-NCP complex were improved to 3.1 Å. Some weak density appeared at a low contour level in the opposite side in this map, indicating there are particles with two motors binding to

the nucleosome. Therefore we further classified the 1,102,628 particles. After focused 3D classification with a soft mask in the opposite motor side, 112,012 particles with two motors (double-binding mode) were selected and refined to 3.3 Å. Another 588,476 particles with one motor were selected and refined to 3.0 Å. The initial movie stacks were finally reprocessed without binning by Relion's Motion Correction program and Bayesian polishing, resulting in an improved overall resolution of 2.8 Å for the map with only one motor. After focused refinement with a soft mask, the NCP part bound with the linker reached 2.8 Å and the motor part was reconstructed to a resolution of 3.1 Å.

The initial model was built by fitting the maps in Chimera using the crystal structures of the individual lobes of ALC1 as determined above and the structure of the NCP from the ISWI-NCP complex (PDB code 6JYL)[29]. The DNA register cannot be unambiguously assigned. Because of the pseudo-symmetric nature of the nucleosome, ALC1 can bind both sides. Since the base identity was not the focus in current study, we did not pursue to assign the orientation of the sequence and used the structure bound by ISWI as the reference. The atomic models for the rest of the molecules were built manually in Coot. The structures were refined using Phenix with secondary structure constrains.

**Reporting summary**. Further information on experimental design is available in the Nature Research Reporting Summary linked to this paper.

## Data availability

Density maps are deposited at the Electron Microscopy Database under accession code EMD-31217, and protein coordinates are deposited at the Protein Data Bank under accession code PDB 7ENN and 7EPU for the nucleosome-bound complex and autoinhibited ALC1, respectively. Source data are provided with this paper. Other data are available from the corresponding author upon reasonable request. Source data are provided with this paper.

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

## Acknowledgements

We thank the Tsinghua University Branch of the China National Center for Protein Sciences (Beijing) for the cryo-EM facility. This work was supported by the National Key Research and Development Program (2019YFA0508902 and 2017YFA0102900 to Z.C.), the National Natural Science Foundation of China (31825016 and 31630046 to Z.C.) and, Advanced Innovation Center for Structural Biology, Tsinghua-Peking Joint Center for Life Sciences.

## Author contributions

L.W. prepared the sample and performed the biochemical analysis; K.C. performed the EM analysis. Z.C. wrote the manuscript with help from all authors; Z.C. directed and supervised all of the research.

## Competing interests

The authors declare no competing interests.
