## [Peer Review File · Nature Communications]

REVIEWER COMMENTS

Reviewer #1 (Remarks to the Author):

Wang et al. present the crystal structure of the chromatin remodeler ALC1/CHD1L from *H. sapiens* bound to a single chain variable fragment (scFv) at a resolution of 3.5 Å and clarify the molecular mechanism of auto-inhibition by the macro domain of ALC1. Additionally, they report the cryo-EM structure of ALC1 bound to a mononucleosome at a resolution of 3.1 Å in the presence of the transition state analog ADP·BeFx. The authors discover a novel interaction between an ALC1 region, located between ATPase lobe 2 and the macro domain, and the acidic patch of the H2A-H2B dimer. Furthermore, the authors conduct biochemical experiments to characterize known cancer mutations of ALC1.

Overall, Wang et al. structurally and biochemically characterize the chromatin remodeler ALC1/CHD1L. The authors present their findings in a concise manner and the manuscript is well written. The cryo-EM data analysis, however, is not sufficiently explained and shows significant inconsistencies that do not meet current standards in the field. Additional biochemical data could additionally support the findings made in this manuscript. Therefore, I support publication if the authors are able to address the concerns below.

Concerns

Overall presentation

1. The authors should give an in-graphic figure legend for all graphs to indicate what colors correspond to which mutant (Fig. 2-4, Suppl. Fig. 7). This was already done for Suppl. Fig. 8b but should be extended to all graphs in this manuscript.
2. To facilitate data interpretation, the authors should make use of additional labels (e.g. in Fig. 3) to clarify the identity of the colored elements. This is not always clear. Using labels that are in the same color as the colored elements also facilitates readability.
3. Use of “ADP·BeF3” vs. “ADP·BeFx” is inconsistent in the manuscript.
4. The authors report resolutions for their cryo-EM with two decimal places. We encourage the authors to report the resolution to only one decimal place. Resolution estimates are not reliable enough to claim resolution beyond one decimal place.
5. In previous publications, the Chen lab successfully employed the use of a DNA-protein interaction scheme (Liu et al, 2017, Nature, Fig 3b, c) to visualize contacts between ATPase motors and the DNA. This is a good resource and reference. The authors should generate a similar figure for this manuscript.
6. To further increase relevancy of the findings for the chromatin remodeling field, the authors should discuss how the auto inhibitory binding of the macro domain 6. compares and contrasts with other regulatory domains in other chromatin remodelers such as the double chromodomain in Chd1.

Crystal structure

1. Presentation of the ALC1 crystal structure without displaying the bound scFv fragment in Fig. 1 conceals the fact that the scFv fragment is also present in this particular crystal structure. The scFv fragment should be shown in Fig. 1 (as already done in Suppl. Fig. 1). This would clarify that the binding of the scFv fragment has no direct impact on the conformation/nature of the ATPase lobe 2/macro domain interface which is critical for the inhibition mechanism of ALC1. The authors should

also explicitly state in the text that the binding of the antibody fragment has no impact on this particular interface.

2. B-factor for the bound ligand is not reported in Supplementary Table 1.

3. The authors report a relatively low percentage for amino acids with favored Ramachandran angles (~86 %). We encourage the authors to improve this quality indicator by refining the structure with tighter geometry restraints.

Biochemistry

1. Generally, the gold standard for measuring ATPase activities is an assay that allows for the regeneration of ATP (e.g. NADH-coupled ATPase assay). The authors employ an assay that monitors the generation of inorganic phosphate through the conversion of an MESG substrate. This assay does not regenerate ATP, which leads to substrate depletion effects. Therefore, the authors should employ an NADH-coupled ATPase assay if enzyme activity is sufficient.

2. Because of the importance of the ALC1 contact with the acidic patch for this manuscript, we also encourage the authors to conduct additional biochemical assays to determine the binding affinity of the linker region with the H2A-H2B dimer using competition assays with a LANA peptide, fluorescent anisotropy or ITC experiments to further validate this interaction. This will be an important resource for the field to understand how chromatin remodelers interact with the acidic patch.

3. Fig. 2d: The figure legend does not explicitly state the time point at which the ATPase activity was determined.

4. Fig. 2d: The authors only demonstrate different levels of ATPase activity on a linear DNA. They should repeat these experiments employing the relevant (i.e. nucleosomal) substrate.

Cryo-EM

The overall presentation of the cryo-EM analysis of the ALC1-nucleosome complex must be significantly improved:

1. Supplementary Fig. 4a, b: Missing scale bar

2. Supplementary Fig. 4b: Could the authors comment on the fact that some of the 2D classes appear to have a second ALC1 remodeler bound to the opposite nucleosome site?

3. Supplementary Fig. 4c: Add label to refinements that indicates obtained resolution

4. Supplementary Fig. 4c: Use thousand separators (commas) to facilitate reading of numbers

5. Supplementary Fig. 4c: Please indicate precise number of initial particles and precise number of particles used in final refinement

6. The number of over one million particles used in the final refinement is relatively high. Did the authors try to further classify the dataset to improve resolution or better resolve certain parts of the map such as the linker-acidic patch interaction? If the authors haven't done this, they must further refine the data to improve the resolution or resolve ALC1 elements better, or alternatively show that further classification does not result in a feature or resolution improvement.

7. It is not clear if the authors employed Bayesian polishing as implemented in RELION to improve the quality of their reconstruction. This is state-of-the-art in the field and should be performed. This procedure will also improve the overall quality of the map and its resolution.

8. Supplementary Fig. 4d: Please indicate resolution where FSC 0.143 threshold is crossed. We

assume that the black horizontal line indicates an FSC of 0.143 but this is also not indicated on the y-axis.

9. Supplementary Fig. 4d: Please include the FSC for map-to-model as implemented in PHENIX.

10. Supplementary Fig. 4d: Please expand the y-axis to include values below 0 as the FSC curves are likely to drop below this value occasionally.

11. Supplementary Fig. 4f: The color legend and the figure legend both do not indicate the identity and unit of the values shown in the surface coloring. We assume it is "resolution in Å" but this is not clear.

12. The authors report an overall resolution of the structure as well as resolutions for the NCP alone and for the motor alone. How were the resolutions for the nucleosome alone and the motor alone obtained? Were the nucleosome and the motor further refined or classified by using masks? Or were the nucleosome and the motor just masked for post-processing? This is not clear and must be clarified in the methods section as well as in the classification tree.

13. Supplementary Fig. 5: The authors do not sufficiently demonstrate the high quality of their data and also do not use the same color scheme for domain coloring as employed in Fig. 2. Therefore the authors should add additional panels that should at least encompass the following elements with density:

Histone octamer

Linker with acidic patch

ATPase motor

Active site of ALC1 with ADP·BeF₃

H4 tail in context of lobe 2

DNA bases at binding site of ALC1 motor domains

14. The pseudo-symmetric nature of the used DNA construct with 10 bp of overlap on each side of the Widom 601 sequence could make a precise assignment of the DNA register difficult. Can the authors unambiguously assign the DNA register based on density? If yes, they should provide a panel in Supplementary Fig. 5 that shows DNA base densities. This would clarify how the authors were able to assign the orientation of the sequence. If the DNA register cannot be unambiguously assigned, the authors should clarify this in the methods section.

15. The description of the cryo-EM data analysis is not consistent and the authors must clarify the used pixel sizes. The original pixel size for data collection of the data set is 0.54125 Å/pixel (again, significant figures with two decimal places should be used). The authors report their final reconstruction with a pixel size of 1.0825 Å/pixel according to the classification tree (cf. Suppl. Fig. 4). The methods section claims a complete unbinning for the final reconstruction. If data was collected in super-resolution mode, this also has to be explicitly stated in the methods section.

16. Raw data of all micrographs and particle positions should be deposited with EMPIAR. Atomic coordinates and corresponding maps (including half maps, masks used for post-processing, final refinement maps, post-processed maps etc. of all maps reported in Supplementary Table 2) must be deposited with the PDB and EMDB.

Reviewer #2 (Remarks to the Author):

ALC1 is a chromatin remodeler required for normal cellular response for DNA repair, and is disrupted in hepatocellular and other cancers. ALC1 has a macrodomain and is known to bind to poly-ADP ribose chains, which stimulates nucleosome sliding activity. Several extensive biochemical and biophysical studies have previously shown that the macrodomain binds to the ATPase motor and that this interaction is inhibitory. Residues involved in the macro-ATPase interaction correlate to cancer-

specific mutations, and these same mutations have been shown to disrupt the macro-ATPase interaction. Further, the binding of PAR to the macrodomain stimulates remodeler activation. In this manuscript, the authors present two novel structures: the ATPase of ALC1 in complex with the macrodomain, and the ATPase and linker bound to a nucleosome. These structures further advance understanding of this interesting remodeling enzyme.

Although these two structures are interesting and informative, the authors failed to put their findings into context of what is already known. In several key aspects, the way the information is framed gives the impression that the authors discovered more than they did. Whether on purpose or not, this is unacceptable. The manuscript therefore requires completely reframing to put all key points into context. If we evaluate what new information is actually given in light of what was previously expected or known, then for a high profile journal I would expect additional experiments to make this story more high impact, rather than just being another structure of an inhibited enzyme.

Major points:

1. The authors cite two recent papers that have provided important prior information and ideas about how ALC1 is regulated and activated: Singh et al., 2017 and Lehmann et al., 2017, back-to-back in Mol Cell. However, the authors fail to indicate key data and hypotheses that these studies gave. These studies reported the following key findings, which need to be articulated clearly and properly attributed:

- (a) the macrodomains bind strongly to the ATPase motor (in trans, as separate molecules);
- (b) acidic residues on ATPase lobe 2 were shown to be critical for interaction with the macrodomain (e.g. macrodomain binding was disrupted with E332A, E334A, D340A, and D353A, shown by Lehmann et al., 2017 Figure S6);
- (c) basic residues on the macrodomain were shown to be needed to bind to the ATPase motor (R857Q, R860W), which were previously shown to be cancer-related;
- (d) binding to polyADP-ribose (PAR) chains releases the macrodomains from the ATPase motor; Singh 2017: "PAR binding to the macrodomain is coupled to the loss of interaction with the ATPase module, consistent with a direct, allosteric regulation of ALC1 by PAR."; Lehmann 2017: "We propose that binding of PARylated PARP1 to the macro domain displaces it from the ATPase of ALC1{complex}";
- (e) the macrodomain is autoinhibitory; Singh 2017: "tri-ADP-ribose binding to the ALC1 ATPase-macrodomain complex strongly activates the ATPase activity of the ALC1 remodeler."

As it presently is given, none of this information is clearly relayed to the reader. Instead, there is an inference that the structure has revealed everything. While it is true that the crystal structure gives details of the interface, it is remarkable how close these other researchers got (demonstrating residues involved in the interface) without structural data. Both studies did HDX and Lehmann did XLMS to identify interacting residues. Lehmann et al. also proposed very clearly that the macrodomains directly bind to ATPase lobe 2 and presented a structural model that is consistent with the crystal structure. Nowhere is this clearly mentioned.

2. The authors show that disruption of macro-ATPase interface increases DNA-stimulated ATPase rate (Fig 2d). Note that Lehmann et al., 2017, gave a similar ATPase plot in their Fig 7a, titled "Figure 7. Mutations that Destabilize ATPase Motor-Macro Domain Interactions Constitutively Activate ALC1 and Alter the Dynamics of Its Recruitment at DNA Damage Sites"

3. The Chen group has had a number of highly impactful studies of chromatin remodelers. One in particular is the ISWI crystal structure showing how the H4 tail binds to ATPase lobe 2. Once again, the authors need to cite their own paper and make it clear to the reader what was known before – that ISWI, CHD1 and SWI/SNF all bind to the H4 tail using the same acidic residues. It is therefore inappropriate to say (page 7): "Consistent with our hypothesis, the acidic pocket of lobe 2 formed by Glu332, Asp377, Asp381 and Asp384 interacts with the basic patch of {16-KRHRK-20} of the H4 tail of the nucleosome." This is presented as though this finding is a completely new and exciting

discovery, when in fact it is exactly what was predicted from the Yan et al., 2016 Nature paper.

4. Another example of where a structural change observed is what was already known (page 7): Likewise, motif IV underwent a loop-to-helix conformational change (Fig. 3e), with the side chain Arg402 binding to the phosphate backbone of nucleosomal DNA." This is already known/expected based on numerous structures.

5. Which mutants are truly novel and unexpected, explained only now by the structure? Were the brace mutants known before? Isn't this the same structure that Swi/Snf ATPases have that Chen group has previously mutated and published in Nature? Was the structure here really critical for understanding these?

6. No rates are given for chromatin remodeling activity. Are these exponential fits (Figure 3), or linear fits to initial slope? Importantly, all of the mutants fail to reach the same amount of remodeled nucleosome as WT, which by itself will make the mutants slower, if the initial slope is used to calculate rates. The authors need to justify how they calculate the rates, and if using initial slope, why it doesn't matter that mutants fail to reach the WT level.

7. The authors state that there is a ~3-fold difference in the rates for wt and D381A. But what is more interesting to ask is what is the source of the defect? Is it just binding? It would be informative to do a Km/kcat analysis by performing the experiment at multiple enzyme/substrate concentrations. This is a potentially interesting mutation, because it is expected to both disrupt H4 tail binding (negative) but also release the macrodomain (positive). As it is, the authors provide data but fail to connect in any meaningful way to the model.

8. A novel finding of this study is that disruption of the Arginines involved in acidic patch binding reduce remodeling activity (R611E and R624E, Figure 3c). The authors need to do the reciprocal experiment, showing that acidic patch mutation of the nucleosome gives a similar defect, and that these mutations are not worse on such acidic-patch mutant nucleosomes. On a related note, why might the R611 mutation be oncogenic? Other mutations are activating yet this shows lower activity, therefore it is unclear how disruption of this interaction is related to the in vivo effects.

9. The W852C mutation increased DNA-stimulated ATPase activity. Does it also hyperactivate remodeling?

10. The final paragraph of the Discussion does not add much.

Minor:

In suppl Fig S9, please also give the motifs that are described in the figures (P-loop, Motif IV, etc). Also, point out Arg in acidic patch binding.

"Ground state" might be better described as "nucleosome free".

Reviewer #3 (Remarks to the Author):

In this paper Wang et al. present the structure of ALC1 in the inhibited state and bound to the nucleosome. They show how the macro domain of ALC1 blocks the H4 tail binding site on lobe 2 of the ATPase and also lowers ATPase activity. In the nucleosome bound state they find that like other SNF-type ATPases the protein bind nucleosomes at the SHL +2 position and interacts with the H4 tail. In addition, the linker domain, between lobe B and the macro domain, interacts with the acidic patch of the nucleosome.

The authors are also able to map cancer mutation on to the structure and helping to rationalize what effects they would have. These mutations they find are located in regions important for DNA binding,

ATP hydrolysis and at domain interfaces. They show that all of these mutations cause a decrease in activity over WT.

We recommend publishing this work. We have a few suggestions below to improve the work.

- Figure 2D, 3C, 4E, 4F and Supp7. label what mutations each curve represents in the figure
- Suppl Fig 2. show side chains
- Suppl Fig 6. show density for nucleotide
- In figure 2 PARP1 is accidentally written as PAPR1

Yuan He

Reviewer #1 (Remarks to the Author):

*Wang et al. present the crystal structure of the chromatin remodeler ALC1/CHD1L from *H. sapiens* bound to a single chain variable fragment (scFv) at a resolution of 3.5 Å and clarify the molecular mechanism of auto-inhibition by the macro domain of ALC1. Additionally, they report the cryo-EM structure of ALC1 bound to a mononucleosome at a resolution of 3.1 Å in the presence of the transition state analog ADP·BeFx. The authors discover a novel interaction between an ALC1 region, located between ATPase lobe 2 and the macro domain, and the acidic patch of the H2A-H2B dimer. Furthermore, the authors conduct biochemical experiments to characterize known cancer mutations of ALC1.*

Overall, Wang et al. structurally and biochemically characterize the chromatin remodeler ALC1/CHD1L. The authors present their findings in a concise manner and the manuscript is well written. The cryo-EM data analysis, however, is not sufficiently explained and shows significant inconsistencies that do not meet current standards in the field. Additional biochemical data could additionally support the findings made in this manuscript. Therefore, I support publication if the authors are able to address the concerns below.

Concerns

Overall presentation

1. The authors should give an in-graphic figure legend for all graphs to indicate what colors correspond to which mutant (Fig. 2-4, Suppl. Fig. 7). This was already done for Suppl. Fig. 8b but should be extended to all graphs in this manuscript.

As suggested, in-graphic figure legends are added in the revised manuscript.

2. To facilitate data interpretation, the authors should make use of additional labels (e.g. in Fig. 3) to clarify the identity of the colored elements. This is not always clear. Using labels that are in the same color as the colored elements also facilitates readability.

Additional labels are added in the revised manuscript

3. Use of “ADP·BeF3” vs. “ADP·BeFx” is inconsistent in the manuscript.

We use ADP·BeFx consistently in the revised manuscript

4. The authors report resolutions for their cryo-EM with two decimal places. We encourage the authors to report the resolution to only one decimal place. Resolution estimates are not reliable enough to claim resolution beyond one decimal place.

As suggested, we report the resolution to only one decimal place in the revised manuscript

5. In previous publications, the Chen lab successfully employed the use of a DNA-protein interaction scheme (Liu et al, 2017, Nature, Fig 3b, c) to visualize contacts between ATPase motors and the DNA. This is a good resource and reference. The authors should generate a similar figure for this manuscript.

A new Suppl. Fig. 7a is added to illustrate this point.

6. To further increase relevancy of the findings for the chromatin remodeling field, the authors should discuss how the auto inhibitory binding of the macro domain 6. compares and

contrasts with other regulatory domains in other chromatin remodelers such as the double chromodomain in Chd1.

As suggested, we compare and contrast the autoinhibition mechanisms of the chromatin remodelers, including Snf2, Chd1 and ISWI, in the discussion section of the revised manuscript.

Crystal structure

1. Presentation of the ALC1 crystal structure without displaying the bound scFv fragment in Fig. 1 conceals the fact that the scFv fragment is also present in this particular crystal structure. The scFv fragment should be shown in Fig. 1 (as already done in Suppl. Fig. 1). This would clarify that the binding of the scFv fragment has no direct impact on the conformation/nature of the ATPase lobe 2/macro domain interface which is critical for the inhibition mechanism of ALC1. The authors should also explicitly state in the text that the binding of the antibody fragment has no impact on this particular interface.

In the revised manuscript, we make a new Fig. 1b to show the position of the scFv, and state explicitly that the antibody has no interaction with the key autoinhibitory macro-lobe 2 interface.

2. B-factor for the bound ligand is not reported in Supplementary Table 1.

B-factor for the bound ligand is added in the Supp. Table 1 in the revised manuscript.

3. The authors report a relatively low percentage for amino acids with favored Ramachandran angles (~86 %). We encourage the authors to improve this quality indicator by refining the structure with tighter geometry restraints.

In the revised manuscript, we perform additional refinements with tighter geometry restraints, and the structure is refined with ~90% favored Ramachandran angles, and 0.1% outlier.

Biochemistry

1. Generally, the gold standard for measuring ATPase activities is an assay that allows for the regeneration of ATP (e.g. NADH-coupled ATPase assay). The authors employ an assay that monitors the generation of inorganic phosphate through the conversion of an MESG substrate. This assay does not regenerate ATP, which leads to substrate depletion effects. Therefore, the authors should employ an NADH-coupled ATPase assay if enzyme activity is sufficient.

In the MESG-based ATPase assays, we used 3 mM ATP, which is in excess over the enzyme (0.1 μ M) used. The reactions led to substrate depletion at the very end of the reactions, but we did not take these data points. Instead, we took the early time points, the linear fractions of the reactions, to compare the activities of varied proteins. To illustrate this point, several cases, WT and three auto-inhibition mutant ALC1 in the presence of DNA, are shown at Suppl. Fig.4 of the revised manuscript. Under these conditions, the ATP substrate was not depleted before the time point of 120s. The MESG-based ATPase assays are very commonly used, and we successfully applied these assays to get sensible measurements of the ATPase activity (Xia, NSMB 2016; Yan, Nature 2016; Liu, Nature 2017). So, we believe the MESG-based ATPase assays are good enough for the measurement of ALC1 ATPase activity shown in this manuscript. We clarify this point in the method section of the revised manuscript. All the raw data of the ATPase assays are provided in the revised manuscript.

2. Because of the importance of the ALC1 contact with the acidic patch for this manuscript, we also encourage the authors to conduct additional biochemical assays to determine the binding affinity of the linker region with the H2A-H2B dimer using competition assays with a LANA peptide, fluorescent anisotropy or ITC experiments to further validate this interaction.

This will be an important resource for the field to understand how chromatin remodelers interact with the acidic patch.

We use MST assays to determine the binding affinity of the linker region with the H2A-H2B dimer. The new data are shown in Fig. 4c in the revised manuscript. Whereas the WT peptide binds to the H2A-H2B dimer with a disassociation constant $K_d \sim 3.7 \mu\text{M}$, the R611E and R614E mutations reduce the affinity over 10-fold, with K_d s $\sim 41 \mu\text{M}$ and $113 \mu\text{M}$, respectively.

3. Fig. 2d: The figure legend does not explicitly state the time point at which the ATPase activity was determined.

The exact time points may vary, depending on the activities of the proteins measured. We clarify this point in the method section of the revised manuscript by stating that “to ensure the ATP substrate was not depleted, the early time points, the linear fractions of the reactions, were used to compare the activities of varied proteins”.

4. Fig. 2d: The authors only demonstrate different levels of ATPase activity on a linear DNA. They should repeat these experiments employing the relevant (i.e. nucleosomal) substrate.

As suggested, we performed the ATPase activities using nucleosome core particle (NCP), in the presence and absence of PARP1, and the data are shown in Fig. 2d in the revised manuscript. Basically, the results obtained from the nucleosome substrate are parallel to those using DNA, in that nucleosome and DNA activated the WT enzyme, which was further enhanced by PARP1. In contrast, the autoinhibitory mutants, D381A, W852C and ALC1(1-673), showed higher basal activities, and were activated by nucleosome and DNA, but no dramatic enhancement occurred in the presence of PARP1.

Cryo-EM

The overall presentation of the cryo-EM analysis of the ALC1-nucleosome complex must be significantly improved:

1. Supplementary Fig. 4a, b: Missing scale bar

Scale bars are added in the revised manuscript.

2. Supplementary Fig. 4b: Could the authors comment on the fact that some of the 2D classes appear to have a second ALC1 remodeler bound to the opposite nucleosome site?

In the revised manuscript, we reconstruct the ALC1 double-binding model, and comment that some nucleosomes were bound by two copies of ALC1 symmetrically, the structure of which was refined to 3.3 \AA (Suppl. Figs 5c, 6 and 7d). In this double-binding mode, ALC1 interacted with the nucleosome in a manner similar to that of the high resolution single-binding structure (Suppl. Fig. 7e), which is the focus of this study.

3. Supplementary Fig. 4c: Add label to refinements that indicates obtained resolution

We add labels to refinements to indicate the obtained resolution in the revised manuscript.

4. Supplementary Fig. 4c: Use thousand separators (commas) to facilitate reading of numbers

We use thousand separators to facilitate reading of numbers of the particles used in the revised manuscript.

5. Supplementary Fig. 4c: Please indicate precise number of initial particles and precise number of particles used in final refinement

We indicate precise numbers of the initial particles and precise number of particles used in final refinement in the revised manuscript.

6. The number of over one million particles used in the final refinement is relatively high. Did the authors try to further classify the dataset to improve resolution or better resolve certain parts of the map such as the linker-acidic patch interaction? If the authors haven't done this, they must further refine the data to improve the resolution or resolve ALC1 elements better, or alternatively show that further classification does not result in a feature or resolution improvement.

Thanks very much for the suggestion. In the revised manuscript, we performed additional classifications, which led to two major classes, one showing the canonical binding mode and the other having two copies of ALC1 bound to the same nucleosome. After Bayesian polishing, the resolution of the canonical structure is improved to 2.8 Å. The data were showed in Suppl. Fig. 5c in the revised manuscript.

7. It is not clear if the authors employed Bayesian polishing as implemented in RELION to improve the quality of their reconstruction. This is state-of-the-art in the field and should be performed. This procedure will also improve the overall quality of the map and its resolution.

In the revised manuscript, we perform Bayesian polishing. In combination of additional classification, we improve the resolution of the map to 2.8 Å.

8. Supplementary Fig. 4d: Please indicate resolution where FSC 0.143 threshold is crossed. We assume that the black horizontal line indicates an FSC of 0.143 but this is also not indicated on the y-axis.

We indicate resolution where FSC 0.143 threshold is crossed in the revised manuscript.

9. Supplementary Fig. 4d: Please include the FSC for map-to-model as implemented in PHENIX.

We include the FSC for map-to-model in the revised manuscript.

10. Supplementary Fig. 4d: Please expand the y-axis to include values below 0 as the FSC curves are likely to drop below this value occasionally.

We expand the y-axis to include values below 0 in the revised manuscript.

11. Supplementary Fig. 4f: The color legend and the figure legend both do not indicate the identity and unit of the values shown in the surface coloring. We assume it is "resolution in Å" but this is not clear.

Yes, the unit is "Å", and we show it in the revised manuscript.

12. The authors report an overall resolution of the structure as well as resolutions for the NCP alone and for the motor alone. How were the resolutions for the nucleosome alone and the motor alone obtained? Were the nucleosome and the motor further refined or classified

by using masks? Or were the nucleosome and the motor just masked for post-processing? This is not clear and must be clarified in the methods section as well as in the classification tree.

The nucleosome and the motor are masked for post-processing. We clarify this point in the methods section as well as in the classification tree in the revised manuscript.

13. Supplementary Fig. 5: The authors do not sufficiently demonstrate the high quality of their data and also do not use the same color scheme for domain coloring as employed in Fig. 2. Therefore the authors should add additional panels that should at least encompass the following elements with density:

Histone octamer

Linker with acidic patch

ATPase motor

Active site of ALC1 with ADP·BeF₃

H4 tail in context of lobe 2

DNA bases at binding site of ALC1 motor domains

We add additional panels to show these elements in the Suppl. Fig. 6 in the revised manuscript.

14. The pseudo-symmetric nature of the used DNA construct with 10 bp of overlap on each side of the Widom 601 sequence could make a precise assignment of the DNA register difficult. Can the authors unambiguously assign the DNA register based on density? If yes, they should provide a panel in Supplementary Fig. 5 that shows DNA base densities. This would clarify how the authors were able to assign the orientation of the sequence. If the DNA register cannot be unambiguously assigned, the authors should clarify this in the methods section.

The DNA register cannot be unambiguously assigned. Because of the pseudo-symmetric nature of the nucleosome, ALC1 can bind both sides, making a precise assignment of the DNA register more difficult. Since the base identity is not the focus in current study, we do not pursue to assign the orientation of the sequence and used the structure bound by ISWI as the reference. We clarify this in the methods section in the revised manuscript.

15. The description of the cryo-EM data analysis is not consistent and the authors must clarify the used pixel sizes. The original pixel size for data collection of the data set is 0.54125 Å/pixel (again, significant figures with two decimal places should be used). The authors report their final reconstruction with a pixel size of 1.0825 Å/pixel according to the classification tree (cf. Suppl. Fig. 4). The methods section claims a complete unbinning for the final reconstruction. If data was collected in super-resolution mode, this also has to be explicitly stated in the methods section.

The data was collected in super-resolution mode, and we clarify this in the revised manuscript. The used pixel sizes are also indicated.

16. Raw data of all micrographs and particle positions should be deposited with EMPIAR. Atomic coordinates and corresponding maps (including half maps, masks used for post-processing, final refinement maps, post-processed maps etc. of all maps reported in Supplementary Table 2) must be deposited with the PDB and EMDB.

PDB-7EPU (crystal structure of ALC1); PDB-7ENN and EMD-31217 for the coordinates and maps

(half-map, masks, final refinement and post-processed maps) of the cryoEM structure of ALC1 bound to the nucleosome. The coordinates and maps will be released upon accepted for publication.

Reviewer #2 (Remarks to the Author):

ALC1 is a chromatin remodeler required for normal cellular response for DNA repair, and is disrupted in hepatocellular and other cancers. ALC1 has a macrodomain and is known to bind to poly-ADP ribose chains, which stimulates nucleosome sliding activity. Several extensive biochemical and biophysical studies have previously shown that the macrodomain binds to the ATPase motor and that this interaction is inhibitory. Residues involved in the macro-ATPase interaction correlate to cancer-specific mutations, and these same mutations have been shown to disrupt the macro-ATPase interaction. Further, the binding of PAR to the macrodomain stimulates remodeler activation. In this manuscript, the authors present two novel structures: the ATPase of ALC1 in complex with the macrodomain, and the ATPase and linker bound to a nucleosome. These structures further advance understanding of this interesting remodeling enzyme.

Although these two structures are interesting and informative, the authors failed to put their findings into context of what is already known. In several key aspects, the way the information is framed gives the impression that the authors discovered more than they did. Whether on purpose or not, this is unacceptable. The manuscript therefore requires completely reframing to put all key points into context. If we evaluate what new information is actually given in light of what was previously expected or known, then for a high profile journal I would expect additional experiments to make this story more high impact, rather than just being another structure of an inhibited enzyme.

Major points:

1. The authors cite two recent papers that have provided important prior information and ideas about how ALC1 is regulated and activated: Singh et al., 2017 and Lehmann et al., 2017, back-to-back in Mol Cell. However, the authors fail to indicate key data and hypotheses that these studies gave. These studies reported the following key findings, which need to be articulated clearly and properly attributed:

(a) the macrodomains bind strongly to the ATPase motor (in trans, as separate molecules);
(b) acidic residues on ATPase lobe 2 were shown to be critical for interaction with the macrodomain (e.g. macrodomain binding was disrupted with E332A, E334A, D340A, and D353A, shown by Lehmann et al., 2017 Figure S6);
(c) basic residues on the macrodomain were shown to be needed to bind to the ATPase motor (R857Q, R860W), which were previously shown to be cancer-related;
(d) binding to polyADP-ribose (PAR) chains releases the macrodomains from the ATPase motor; Singh 2017: "PAR binding to the macrodomain is coupled to the loss of interaction with the ATPase module, consistent with a direct, allosteric regulation of ALC1 by PAR."; Lehmann 2017: "We propose that binding of PARylated PARP1 to the macro domain displaces it from the ATPase of ALC1{complex}";
(e) the macrodomain is autoinhibitory; Singh 2017: "tri-ADP-ribose binding to the ALC1 ATPase-macrodomain complex strongly activates the ATPase activity of the ALC1 remodeler."

As it presently is given, none of this information is clearly relayed to the reader. Instead, there is an inference that the structure has revealed everything. While it is true that the crystal structure gives details of the interface, it is remarkable how close these other

researchers got (demonstrating residues involved in the interface) without structural data. Both studies did HDX and Lehmann did XLMS to identify interacting residues. Lehmann et al. also proposed very clearly that the macrodomains directly bind to ATPase lobe 2 and presented a structural model that is consistent with the crystal structure. Nowhere is this clearly mentioned.

In the previous submission, the introduction section contained < 400 words due to the limitation of the total length. Nevertheless, we cited the two key reference papers by Singh et al., 2017 and Lehmann et al., 2017, and included the two most important points of ALC1 regulation: autoinhibition by the macro domain, and activation by the binding of the PAR chain to the macro domain, without going too many biochemical details.

In the revised manuscript, we include one paragraph to summarize these early studies of ALC1 regulation.

2. The authors show that disruption of macro-ATPase interface increases DNA-stimulated ATPase rate (Fig 2d). Note that Lehmann et al., 2017, gave a similar ATPase plot in their Fig 7a, titled "Figure 7. Mutations that Destabilize ATPase Motor-Macro Domain Interactions Constitutively Activate ALC1 and Alter the Dynamics of Its Recruitment at DNA Damage Sites"

We noticed that and cited the works by Singh and Lehmann in the previous submission by stating that this structure is also supported by the cancer-associated mutations at Arg857 and Arg860, which release the autoinhibition of ALC1 {Singh, 2017;Lehmann, 2017}.

3. The Chen group has had a number of highly impactful studies of chromatin remodelers. One in particular is the ISWI crystal structure showing how the H4 tail binds to ATPase lobe 2. Once again, the authors need to cite their own paper and make it clear to the reader what was known before – that ISWI, CHD1 and SWI/SNF all bind to the H4 tail using the same acidic residues. It is therefore inappropriate to say (page 7): "Consistent with our hypothesis, the acidic pocket of lobe 2 formed by Glu332, Asp377, Asp381 and Asp384 interacts with the basic patch of {16-KRHRK-20} of the H4 tail of the nucleosome." This is presented as though this finding is a completely new and exciting discovery, when in fact it is exactly what was predicted from the Yan et al., 2016 Nature paper.

We believe it is great to have the experimental evidence showing that the acidic pocket of ALC1 binds to the H4 tails. Moreover, it is unexpected that the H4-binding surface is sequestered by the macro domain. These findings provide the structural basis for the mechanisms of ALC1 autoinhibition, and its activation by the H4 tails.

To clarify these points in the revised manuscript, we state that the acidic pocket formed by Glu332, Asp377, and Asp381 of ALC1 is similar to those of ISWI and Snf2 (Suppl. Fig. 1c), which bind to the H4 tails {Yan, 2016; Liu, 2017}. At page 8, we state that in line with the structures of Snf2 and ISWI {Yan, 2016; Liu, 2017}, the acidic pocket of lobe 2 formed by Glu332, Asp377, Asp381, and Asp384 of ALC1 interacts with the basic patch K₁₆R₁₇H₁₈R₁₉ of the H4 tail of the nucleosome. Moreover, at the discussion section, we compare and contrast the regulation mechanisms of Snf2, Chd1, ISWI and ALC1.

4. Another example of where a structural change observed is what was already known (page 7): Likewise, motif IV underwent a loop-to-helix conformational change (Fig. 3e), with the side chain Arg402 binding to the phosphate backbone of nucleosomal DNA." This is already known/expected based on numerous structures.

We simply show the experimental observation that motif IV underwent a loop-to-helix conformational change. We do not claim that the helical conformation of motif IV at the activated state is unexpected. To clarify this point, we state in the revised manuscript that motif IV underwent a loop-to-helix conformational change (Fig. 3e), with the side chain of Arg402 binding to the phosphate backbone of nucleosomal DNA, as typically observed in the activated state of chromatin remodelers {Yan, 2020}.

What is unexpected is that in the autoinhibited state, the macro domain binds motif IV, and induces a loop conformation, sequestering this DNA-binding element. We consider it is valuable to report the activated conformation, showing that release of the macro domain frees the H4-binding surface and motif IV to interact with the nucleosome, providing the mechanism of ALC1 activation.

5. Which mutants are truly novel and unexpected, explained only now by the structure? Were the brace mutants known before? Isn't this the same structure that Swi/Snf ATPases have that Chen group has previously mutated and published in Nature? Was the structure here really critical for understanding these?

There are several novel and unexpected mutants, including Arg611 and Arg614, which bind to the acidic patch of the nucleosome, and the cancer-related W952C mutant, which releases the ALC1 autoinhibition. The R260M mutant is not a brace helix mutant and not known before. It maps to lobe 1 and interacts with the brace helix at the lobe 1-lobe 2 interface at the nucleosome-bound state. We clarify this in the revised manuscript. Likewise, no mutant equivalent to R319Q, which is close to the nucleosomal DNA, was reported or tested before. These mutations dysregulate ALC1 in different aspects. Therefore, we argue that the remodeling activity of ALC1 is regulated through multiple fashions, our findings provide a framework to understand these cancer-related mutations.

6. No rates are given for chromatin remodeling activity. Are these exponential fits (Figure 3), or linear fits to initial slope? Importantly, all of the mutants fail to reach the same amount of remodeled nucleosome as WT, which by itself will make the mutants slower, if the initial slope is used to calculate rates. The authors need to justify how they calculate the rates, and if using initial slope, why it doesn't matter that mutants fail to reach the WT level.

The observation that the mutants fail to reach the same amount of remodeled nucleosome as WT reflects the low remodeling activities of the mutant enzymes. The reasons for the low remodeling product formation are not completely clear. A possibility is inhibition by the product of ATP hydrolysis (ADP). The R260M mutation seems to perturb the thermal stability, as the mutant enzyme remodels a higher amount of the nucleosome at a lower temperature, to a level similar to WT. Since the reactions reach comparable plateau levels, we fit the data and report the remodeling rate constants of the mutant (0.05 min^{-1}) and WT protein (0.1 min^{-1}). The data suggest the R260M mutation impacts both the stability and remodeling efficiency of the enzyme. The new data are shown in Fig. 5h and Suppl. Fig. 9f and 9g in the revised manuscript. The data are exponential fit, and we clarify this in the method section of the revised manuscript.

Thermal stability does not seem to play an important role for the arginine anchor mutants, as reducing the reaction temperature, or adding more enzyme (R611E) at the later time points do not notably enhance the product formation. We clarify this point, and show the new data in Suppl. Fig. 9e of the revised manuscript. Since the reactions catalyzed by the WT and the mutants reached quite different plateau levels, we preferred not to report the rate constant, but qualitatively described the difference.

7. The authors state that there is a ~3-fold difference in the rates for wt and D381A. But what is more interesting to ask is what is the source of the defect? Is it just binding? It would be informative to do a K_m/k_{cat} analysis by performing the experiment at multiple

enzyme/substrate concentrations. This is a potentially interesting mutation, because it is expected to both disrupt H4 tail binding (negative) but also release the macrodomain (positive). As it is, the authors provide data but fail to connect in any meaningful way to the model.

We do the K_m/K_{cat} catalysis and clarify this point in the revised manuscript. The functions of Asp381 are twofold. It interacts with the macro domain in the nucleosome-free state (Fig. 2c), inhibiting the enzyme in the absence of the substrate, whereas it interacts with H4-tail in the nucleosome-bound state, promoting the remodeling activity. To get insights into the loss of the remodeling activity caused by the D381A mutation at the activated state, we performed the experiments at multiple enzyme/substrate concentrations to do K_m/K_{cat} analysis using the construct without the macro domain, which bypassed the complication due to the autoinhibition and requirement for PARP1 activation, Fig. 3c. The data showed that the D381A mutation increased the K_m ~2-fold, while it slightly decreased the catalytic efficacy (K_{max}), suggesting the defect was mainly because of the compromised nucleosome binding.

8. A novel finding of this study is that disruption of the Arginines involved in acidic patch binding reduce remodeling activity (R611E and R624E, Figure 3c). The authors need to do the reciprocal experiment, showing that acidic patch mutation of the nucleosome gives a similar defect, and that these mutations are not worse on such acidic-patch mutant nucleosomes. On a related note, why might the R611 mutation be oncogenic? Other mutations are activating yet this shows lower activity, therefore it is unclear how disruption of this interaction is related to the in vivo effects.

In the revised manuscript, we do the reciprocal experiment with mutations of the acidic patch to further validate these findings. Consistent with our model, disruption of the acidic patch reduced the remodeling activity of the WT enzyme, to a level similar to the activities of the ALC1 mutants towards the WT nucleosome, Fig. 4d. Moreover, the R611E and R614E mutations were not worse on the acidic patch mutant nucleosomes.

Tight regulation of ALC1 action, chromatin remodelers in general, is critical for the function in vivo. We agree that it is unclear how disruption of ALC1 regulation is directly related to the in vivo effects. In the discussion section, we provide some education-based speculations. We clarify this point in the revised manuscript.

9. The W852C mutation increased DNA-stimulated ATPase activity. Does it also hyperactivate remodeling?

In the revised manuscript, we measure the chromatin remodeling activity of W852C, and the data are showed in Fig. 5b. The W852C mutant is hyperactive in the absence of the PARP1 activation, whereas it displays an activity similar to that of the WT protein in the presence of PARP1, suggesting Trp852 is important for autoinhibition of ALC1, but dispensable for the remodeling activity when the enzyme is activated by PARP1 in vitro.

10. The final paragraph of the Discussion does not add much.

The fundamental science of ALC1 biology, such as the structure and the regulation mechanisms, is very interesting. ALC1 is also interested to the medical branch of research. We believe that it will broaden the reader scope of this manuscript by discussing the varied regulation modes of ALC1, and the rich opportunities and a large chemical space to control the function of this enzyme in cells. So, we would like to keep this part of the manuscript.

Minor:

In suppl Fig S9, please also give the motifs that are described in the figures (P-loop, Motif IV, etc). Also, point out Arg in acidic patch binding.

As suggested, more functional elements are annotated in the revised manuscript.

“Ground state” might be better described as “nucleosome free”.

As suggested, “nucleosome-free” state is used.

Reviewer #3 (Remarks to the Author):

In this paper Wang et al. present the structure of ALC1 in the inhibited state and bound to the nucleosome. They show how the macro domain of ALC1 blocks the H4 tail binding site on lobe 2 of the ATPase and also lowers ATPase activity. In the nucleosome bound state they find that like other SNF-type ATPases the protein binds nucleosomes at the SHL +2 position and interacts with the H4 tail. In addition, the linker domain, between lobe B and the macro domain, interacts with the acidic patch of the nucleosome.

The authors are also able to map cancer mutations on to the structure and helping to rationalize what effects they would have. These mutations they find are located in regions important for DNA binding, ATP hydrolysis and at domain interfaces. They show that all of these mutations cause a decrease in activity over WT.

We recommend publishing this work. We have a few suggestions below to improve the work.

- *Figure 2D, 3C, 4E, 4F and Supp7. label what mutations each curve represents in the figure*
As suggested, in-graphic figure legends are added in the revised manuscript.

- *Suppl Fig 2. show side chains*
Side chains are shown as suggested.

- *Suppl Fig 6. show density for nucleotide*
EM density for nucleotide is shown Suppl. Fig. 6 in the revised manuscript.

- *In figure 2 PARP1 is accidentally written as PAPR1*
The typo is corrected in the revised manuscript.

REVIEWER COMMENTS

Reviewer #1 (Remarks to the Author):

Structural basis of ALC1/CHD1L autoinhibition and the mechanism of activation by the nucleosome

Wang et al. present the crystal structure of the chromatin remodeler ALC1/CHD1L from *H. sapiens* bound to a single chain variable fragment (scFv) and report the cryo-EM structure of ALC1 bound to a mononucleosome. The revised version of the manuscript is much clearer with a noticeable improvement in the cryo-EM data processing and the presentation of the structural results. Additionally, the authors provide additional biophysical and biochemical data (Fig. 4c) to better understand the binding interface of the linker region of ALC1 with the nucleosomal acidic patch.

The authors should still improve their presentation by clearly indicating the time points selected for determining the ATPase activity of each proteins used in Fig 2d. This is especially important because the MESH-based ATPase assay is suboptimal with regards to potential substrate depletion. It is therefore very important that the authors transparently reveal what time points were used for the determination of the ATPase activity.

We also encourage the authors to carefully check their manuscript for spelling and grammatical errors that are still sporadically found in the manuscript and create a negative impression during reading. For example:

“We performed the experiments at multiple enzyme/substrate concentrations to a do Km/Kcat analysis using [...]” should be “We performed the experiments at multiple enzyme/substrate concentrations to do a Km/Kcat analysis using [...]”

“The Arg857 and Arg860 mutants were previously showed to be constitutively active.” Should be “The Arg857 and Arg860 mutants were previously shown to be constitutively active.”

“All data points are showed with lines indicating the means (n=3).” should be “All data points are shown with lines indicating the means (n=3).”

Reviewer #2 (Remarks to the Author):

Overall, the revised manuscript is much improved and I support publication. I should mention that there is a recent paper that reported the ALC1 linker bound to the acidic patch of the nucleosome – “Mechanistic Insights into Regulation of the ALC1 Remodeler by the Nucleosome Acidic Patch” Cell Reports 2020 Dec 22;33(12):108529. doi: 10.1016/j.celrep.2020.108529. <https://pubmed.ncbi.nlm.nih.gov/33357431/> This paper needs to also be cited in the introduction and discussion, as several core findings in the manuscript overlap with this paper. While that paper reduces the novelty of the discovery that ALC1 binds to of acidic patch binding, the manuscript here is still an important advance, offering the first high-resolution view of the ALC1 inhibited state. As before, however, it is important to put the findings of the authors in the appropriate context to work of others in the field, so some adjustment of the text to properly cite this recently paper would be the only further change required before publication.

Note that the paper should be checked for proper English grammar. Two examples where changes are needed are

Page 5: “PARP1 did not further activated” should be “PARP1 did not further activate”

Page 11: “sequestrates” should be “sequesters”

Reviewer #1 (Remarks to the Author):

Structural basis of ALC1/CHD1L autoinhibition and the mechanism of activation by the nucleosome

Wang et al. present the crystal structure of the chromatin remodeler ALC1/CHD1L from *H. sapiens* bound to a single chain variable fragment (scFv) and report the cryo-EM structure of ALC1 bound to a mononucleosome. The revised version of the manuscript is much clearer with a noticeable improvement in the cryo-EM data processing and the presentation of the structural results. Additionally, the authors provide additional biophysical and biochemical data (Fig. 4c) to better understand the binding interface of the linker region of ALC1 with the nucleosomal acidic patch.

The authors should still improve their presentation by clearly indicating the time points selected for determining the ATPase activity of each proteins used in Fig 2d. This is especially important because the MESG-based ATPase assay is suboptimal with regards to potential substrate depletion. It is therefore very important that the authors transparently reveal what time points were used for the determination of the ATPase activity.

We indicate the specific time points selected for determination of the ATPase activity in the methods section of this revised manuscript.

We also encourage the authors to carefully check their manuscript for spelling and grammatical errors that are still sporadically found in the manuscript and create a negative impression during reading. For example:

“We performed the experiments at multiple enzyme/substrate concentrations to a do Km/Kcat analysis using [...]” should be “We performed the experiments at multiple enzyme/substrate concentrations to do a Km/Kcat analysis using [...]”

“The Arg857 and Arg860 mutants were previously showed to be constitutively active.”

Should be “The Arg857 and Arg860 mutants were previously shown to be constitutively active.” “All data points are showed with lines indicating the means (n=3).” should be “All data points are shown with lines indicating the means (n=3).”

We corrected these and other errors we found.

Reviewer #2 (Remarks to the Author):

Overall, the revised manuscript is much improved and I support publication. I should mention that there is a recent paper that reported the ALC1 linker bound to the acidic patch of the nucleosome – “Mechanistic Insights into Regulation of the ALC1 Remodeler by the Nucleosome Acidic Patch” *Cell Reports* 2020 Dec 22;33(12):108529. doi: 10.1016/j.celrep.2020.108529. <https://pubmed.ncbi.nlm.nih.gov/33357431/> This paper needs to also be cited in the introduction and discussion, as several core findings in the manuscript overlap with this paper. While that paper reduces the novelty of the

discovery that ALC1 binds to of acidic patch binding, the manuscript here is still an important advance, offering the first high-resolution view of the ALC1 inhibited state. As before, however, it is important to put the findings of the authors in the appropriate context to work of others in the field, so some adjustment of the text to properly cite this recently paper would be the only further change required before publication.

In the discussion of this revised manuscript, we cite and discuss this recent paper by Lehmann et al., which, however, shows an interaction mode different from our structure. The Lehmann's structure revealed only one of the arginine anchors. More importantly, the isolated ALC1 linker peptide bound to the nucleosome with Arg611 interacting Glu64, Glu61, Asp90 and Hlu92 of H2A, whereas Arg614 bound to this acidic pocket in our structure. We noticed that the EM density, which fit neatly to the side chain of Leu613 in our structure, could not easily be explained by the smaller side chain of Ser612 in the Lehmann's structure.

Note that the paper should be checked for proper English grammar. Two examples where changes are needed are

Page 5: "PARP1 did not further activated" should be "PARP1 did not further activate"

Page 11: "sequestrates" should be "sequesters"

We corrected these and other errors we found.